# Responses of intended and unintended receivers to a novel sexual signal suggest clandestine communication

Robin M. Tinghitella [1,3✉], E. Dale Broder [2,3], James H. Gallagher [1], Aaron W. Wikle [1] & David M. Zonana [1]

Inadvertent cues can be refined into signals through coevolution between signalers and receivers, yet the earliest steps in this process remain elusive. In Hawaiian populations of the Pacific field cricket, a new morph producing a novel and incredibly variable song (purring) has spread across islands. Here we characterize the current sexual and natural selection landscape acting on the novel signal by (1) determining fitness advantages of purring through attraction to mates and protection from a prominent deadly natural enemy, and (2) testing alternative hypotheses about the strength and form of selection acting on the novel signal. In field studies, female crickets respond positively to purrs, but eavesdropping parasitoid flies do not, suggesting purring may allow private communication among crickets. Contrary to the sensory bias and preference for novelty hypotheses, preference functions (selective pressure) are nearly flat, driven by extreme inter-individual variation in function shape. Our study offers a rare empirical test of the roles of natural and sexual selection in the earliest stages of signal evolution.

[1] Department of Biological Sciences, University of Denver, Denver, CO, USA. [2] Department of Biology, St Ambrose University, Davenport, IA, USA. [3] These authors contributed equally: Robin M. Tinghitella, E. Dale Broder. ✉email: robin.tinghitella@du.edu

The complexity and diversity of life challenges us to understand how novel forms evolve. While novelty is readily detectable on a macroevolutionary scale, the microevolutionary processes that result in novel traits remain relatively unknown because observing the contemporary evolution of new traits is so very rare[1]. Sexual signals, the focus of this work, are frequently the only or the most divergent traits between populations[2,3], strongly implicating sexual selection in speciation and the generation of biodiversity[4,5]. Yet, how novel sexual signals arise, persist, and spread is difficult to envision because signals and receiver responses frequently coevolve, and new signal features could disrupt existing communication systems. Even the rapid evolution of existing sexual signals appears rare (reviewed in refs. [6,7]), perhaps because evolutionary change in sexual signals is constrained by the coupling of signals with receiver responses, genetic architecture, and integration with complex behavior[6,8]. How then do novel signals come to be?

While we know little about the mechanisms that would favor the maintenance or fixation of a new signal, we know even less about the matching changes that must occur in the receiver to produce a coupled response[9]. To answer questions about the role of sexual selection in diversification and speciation, researchers have necessarily made comparisons between closely related species or populations that have differed in sexual signals for many generations, using phylogenetic approaches to reconstruct the evolution of sexual signals and receiver responses and investigating current mechanisms of reproductive isolation between extant groups (reviewed in ref. [4]). In one recent example, Jewel wasps (*Nasonia vitripennis*) evolved a novel pheromone from an existing compound, and an initially neutral response from female wasps evolved into a preference for the new compound[10]. However, even in this excellent example, because we are looking back on past evolutionary change, reasonable alternative explanations cannot always be tested[4]. To truly understand the evolutionary context for signal origins, we need to address our questions in study systems that are currently experiencing signal evolution[9].

Theoretically, inadvertent cues resulting from mutations or cues previously not associated with mating could be refined into sexual signals through coevolution between males and females, shaped by receiver sensory and cognitive capabilities and pre-existing biases[4,11–13]. For a new putative sexual signal to persist initially, it could either be detectable to receivers and aid in communication or be selectively neutral and not costly, persisting until preferences later evolve[10,14]. Receivers have sensory systems that are broadly tuned such that they perceive and attend to many more aspects of their surroundings than the signals of potential mates[15,16]. Even if a preference is absent when a new signal emerges, as long as receivers can detect the signal, associative learning through positive sexual experiences could quickly generate a new preference (e.g. refs. [17,18]). Receivers also often have preferences for trait values or combinations that are absent in courters, such as supernormal or novel stimuli[4], which may be expressed when exposed to researcher-manipulated signals (e.g. refs. [19–24]). Such hidden preferences originate from pre-existing biases unrelated to the current distribution of courter traits (e.g. stemming from perceptual biases[13] or a prior history of exposure to traits that are currently absent[12]). Like novel signals, these preferences could persist in a population hidden from selection if there is no cost to the receiver for carrying the preference[25].

In addition to being subject to selection imposed by intended receivers (often female conspecifics), sexual signals are quite famously subject to the attention of unintended receivers like predators and parasites that use conspicuous sexual signals to locate potential hosts and prey (reviewed in ref. [26]). Intended and unintended receivers often prefer the same conspicuous components of sexual signals (e.g. ref. [27]), exerting opposing selection pressures on the signal, which can result in stabilizing selection (e.g. ref. [28]) as well as plastic and evolutionary shifts in signaler and receiver behavior (e.g. refs. [29,30]). So when new signals do evolve, we should consider the manner in which the initial variation in those traits is shaped by both conspecific receivers and natural enemies.

Pacific field crickets, *Teleogryllus oceanicus*, are native to Australia, island-hopped through the Pacific, and colonized the Hawaiian Islands between 150 and 2500 years ago[31,32]. The crickets attract female mates from afar using a long-distance calling song that has been described as nearly pure tone with a peak frequency of ~4.8 kHz (Supplementary Fig. 1[33,34]). Recent and dramatic changes to the crickets' sexual signal within Hawaii[8,35] are some of the very few examples of rapid evolution in a sexual signal occurring in natural populations[6,7]. In the early 2000s, a sex-linked mutation (flatwing) spread through a population on the Hawaiian island of Kaua'i over fewer than 20 generations, rendering >95% of males obligately silent, and a similar silencing mutation was found on O'ahu a few years later[8,35,36]. Silent males are protected from a North American parasitoid fly, *Ormia ochracea*, that colonized Hawaii sometime before 1989[37], co-occurs with *T. oceanicus* only in Hawaii[38], and hunts for hosts using the crickets' song[39]. Signal loss was facilitated by dramatically relaxed mating requirements of Hawaiian females relative to females from elsewhere in their range (Australia, Oceania)[40]. The phenotypic composition of Hawaiian populations has fluctuated dynamically since the discovery of the flatwing mutation[41–43].

More recently, we discovered a population of *T. oceanicus* at Kalaupapa National Historical Park on the island of Moloka'i (hereafter Kalaupapa) where males produce a novel purring sound that is spectrally unlike the typical ancestral songs (hereafter, ancestral) using structurally different wing morphology (Fig. 1)[41]. Purring males' songs have higher median peak frequency, lower amplitude, and are more broadband than ancestral songs[41] (Supplementary Fig. 1). And, some lab-born females from Kalaupapa can hear purring and use it to locate males from 1 m away in the lab[41]. Purring does appear to be heritable as it has persisted in common garden over multiple generations, and while the spectral characteristics of song are relatively invariant among ancestral type males, the characteristics of purring songs are incredibly variable among individuals[41] (Fig. 2). Over the last 3 years, our systematic sampling revealed that the purring phenotype is also found in additional populations (several of which have been continuously studied since the early 1990s) that previously contained only silent or silent and ancestral type males[42], resulting in five populations on three islands that now contain purring males (Fig. 1). While we do not know how long purring has been present in Kalaupapa[41], and purring may have existed at very low frequencies for some time in other locations, appreciable numbers of purring males were not present in systematic samples of any Hawaiian population prior to 2017. Other morphs of *T. oceanicus* have also recently been described (e.g. small- and curly-winged morphs[44]), but here we focus on purring songs as first described by Tinghitella et al.[41]. Long-studied populations of *T. oceanicus* across the Hawaiian islands that now contain appreciable numbers of purring crickets, but did not previously, present an unusual opportunity to uncover how the initial stages of novel signal evolution proceed, during which new traits are shaped by choosy conspecifics and eavesdropping natural enemies.

We capitalize on our discovery of a new signal in replicate long-studied populations to examine the responses of female conspecifics and natural enemies to a new sexual signal at time zero (as close to the initial evolution of the trait as possible).

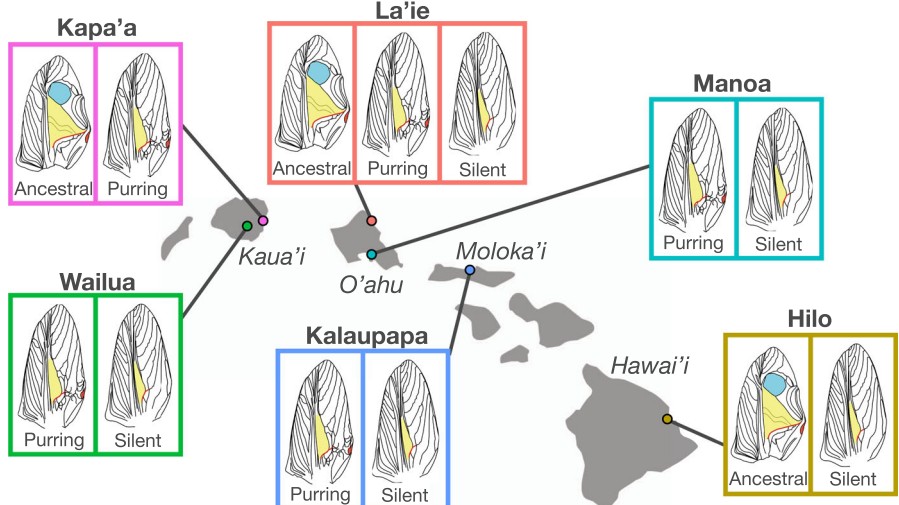

**Fig. 1 Map of the Hawaiian Islands indicating presence of three major cricket morphs (ancestral, purring, and silent) across six sampling populations.** Wing drawings adapted from Fig. 1 in Tinghitella et al. [41].

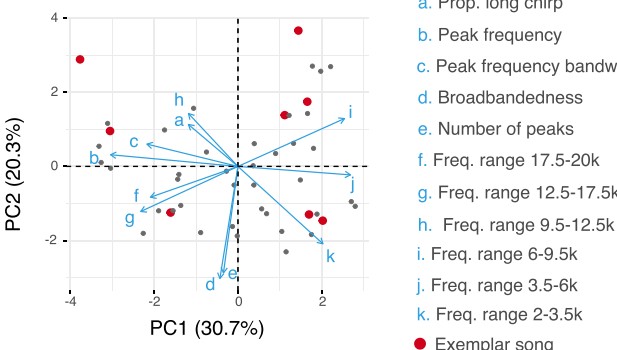

a. Prop. long chirp
b. Peak frequency
c. Peak frequency bandwidth
d. Broadbandedness
e. Number of peaks
f. Freq. range 17.5-20k
g. Freq. range 12.5-17.5k
h. Freq. range 9.5-12.5k
i. Freq. range 6-9.5k
j. Freq. range 3.5-6k
k. Freq. range 2-3.5k
● Exemplar song

**Fig. 2 PCA loadings of purring song characteristics (a–k) including frequency, temporal patterning, and broadbandedness (Table 2).** Points show the 46 songs used in the PCA. Songs chosen for phonotaxis in the exemplar experiment are indicated in red. Source data are provided as a Source Data file.

To directly measure the initial selective environment surrounding a novel animal signal, we test responses of female crickets (sexual selection) and flies (natural selection) from six independently evolving[31,32] Hawaiian populations that contain different ratios and combinations of the ancestral, purring, and silent morphs (Fig. 1) and experience different fly parasitism rates[39,45]. We first address the fitness consequences of purring, hypothesizing that the new morph is more protected from parasitoid flies than the ancestral, and attracts more female mates than the silent morph (i.e., that purring is an alternative evolutionary solution to the conflicting selection pressures acting on male song). If crickets respond positively to purring songs, but flies do not, this would suggest that purring is a private mode of communication amongst crickets.

Next, to test several hypotheses about the selective landscapes that favor the establishment of a novel signal, we generate preference functions by characterizing individual and population-level cricket and fly responses to continuous variation in purring calling song. We do this with two experiments: the first experiment varies only peak frequency, given its importance in song recognition[46–48] (hereafter frequency manipulation experiment), while the second uses purring song exemplars that represent much of the variation in purring songs (hereafter exemplar

experiment). The shape of the resulting preference functions reflects selection acting on purring song variation. Classic literature suggests that new signals evolve within the sensory space of the receivers[49], and initial responses to purring songs may thus reflect the sensory abilities of female crickets and flies. Assuming no recent evolutionary change in cricket or fly hearing ability, this hypothesis predicts that receivers will be most responsive to songs with a peak frequency that matches the ancestral song (~4.8 kHz[41]), which falls within the range of elevated sensory capabilities for both parties[50,51]. A second hypothesis is that crickets prefer novel, unfamiliar stimuli like rare song variants or those with characteristics that differ most from the ancestral (e.g. ref. [52]; reviewed in ref. [4]). A third possibility is that preference functions may be flat, reflecting no particular peak preference close to the origin of purring. Such functions would lend support to the hypothesis that the evolution of peak preference lags behind the evolutionary origin of the purring song itself. Lacking a particular peak preference, however, does not preclude differences in preference function shape; for instance, populations with no peak preference may still differ in responsiveness[53–56]. Explicit comparisons of preference functions across populations that contain different combinations of morphs (some of which have no or few purring males; Hilo and Kapa'a) can reveal differences in selection pressures and evolutionary history, allowing us to make predictions about how the new signal may diverge across the Hawaiian archipelago.

Here, we show that female *T. oceanicus* respond positively to novel purring songs, but parasitoid flies do not, suggesting purring may be a new evolutionary solution to conflicting natural and sexual selection pressures that allows private communication among crickets. In the frequency manipulation and exemplar experiments, preference functions are nearly flat but driven by extreme inter-individual variation in function shape. This finding does not support the sensory bias hypothesis or the preference for novelty hypothesis. We capitalize on a rare opportunity to gather some of the first data illustrating how natural and sexual selection act in the earliest stages of signal evolution.

## Results

**Search behavior in response to ancestral song, purring song, and white noise.** Positive phonotactic behavior in female crickets depended strongly upon song type; the percentage of positive responses in phonotaxis trials to ancestral song was 78.3%, to

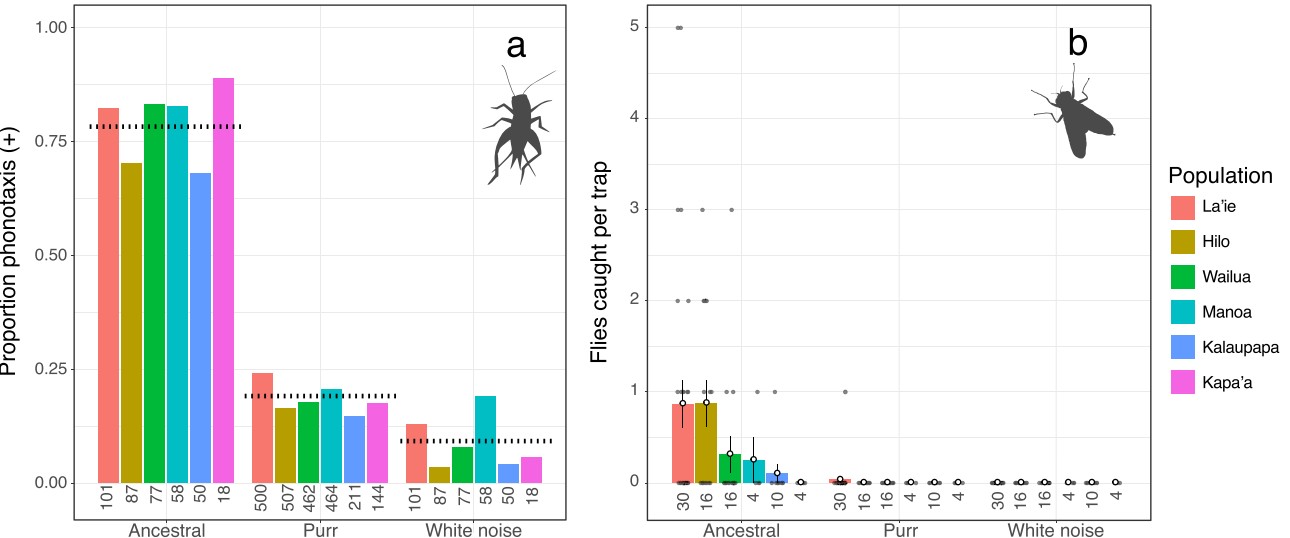

**Fig. 3 Positive phonotactic behavior in female crickets and flies depends upon song type. a** Playbacks of the ancestral calling song elicited greater rates of phonotactic responses by female crickets than playbacks of the derived purring song or white noise control, while female crickets also responded more positively to purring song than to white noise. Data shown here come from phonotaxis responses of crickets in both the frequency and exemplar experiments. Dotted lines indicate the proportion of crickets exhibiting positive phonotaxis averaged across populations. **b** Fly traps playing ancestral song caught the overwhelming majority of flies in the field, with only a single fly caught at a trap playing the purring song, and none caught at traps broadcasting white noise (open points = average number of flies/trap; whiskers = SE; gray points = raw data). Colors of bars indicate populations of origin, with sample sizes indicated below the bars (N = numbers of individual phonotaxis trials and traps deployed for crickets and flies, respectively). Source data are provided as a Source Data file.

purring was 19.1% and to white noise was 9.2% (binomial GLMM: LR $X^2 = 483.07$, df = 2, $p < 0.0001$; Fig. 3a). Crickets responded much more positively to the ancestral song than the derived purring song or the white noise control, and importantly, also responded more positively to purring song than to white noise (Tukey's contrasts of estimated marginal means; Ancestral—Purr: estimate = 3.02, z-ratio = 19.92, $p < 0.0001$; Ancestral—WN: estimate = 3.89, z-ratio = 17.19, $p < 0.001$; Purr—WN: estimate = 0.862, z-ratio = 4.67, $p < 0.0001$). In the field, fly traps broadcasting ancestral song caught 47 female flies (0.59 flies/trap), while we only captured a single fly at traps playing purring song (0.012 flies/trap), and none at white noise traps (Fig. 3b). While this single fly locating a purring speaker could be an anomaly, when we have broadcast only purring song in funnel trap arrays (without competition from nearby ancestral songs), we caught parasitoid flies at a similar rate; three flies were collected at 78 traps (0.038 flies/trap) across the six populations.

**Frequency manipulations: Cricket and fly preferences for peak frequency.** We first visualized the preference functions of female crickets and flies for purring song frequency (Fig. 4). A particularly prominent pattern in the frequency preference functions is the extreme inter-individual variation for both crickets and flies in function direction and shape (faint solid lines in Fig. 4), but nearly flat aggregate preference functions (thick dashed lines in Fig. 4), regardless of which metrics of female response were measured. We found no evidence that female crickets or flies prefer certain purring song frequencies over others, as the frequency of manipulated songs did not predict their responses (coefficients for frequency and frequency$^2$ terms in LMMs and GLMMs do not differ significantly from zero; Supplementary Tables 1 and 2). In other words, preference functions aggregated at the population (crickets) and species-level (flies) appear flat at time zero. However, responses by individual females to frequency were extremely variable (faint solid lines in Fig. 4; see individual: population and individual random effects in Supplementary

Tables 1 and 2), suggesting flat overall functions are driven by extreme inter-individual variation, rather than a consistent lack of preference, per se. Inter-individual variance in female cricket responses was at least twice as great as variance at the population-level (see relative effect sizes of random effects in the LMMs and GLMMs, Supplementary Table 1), and intra-class correlations between observations of the same individual were much stronger than correlations between different individuals within the same population (Supplementary Table 1), suggesting that variation in responses disproportionately occurs among individuals.

Despite the importance of individual-level variation, we also find differences in the phonotactic responses of female crickets from different populations (Table 1), generating subtle variation in the shape of the aggregate preference functions. For example, for phonotaxis, responsiveness (the overall elevation of the preference function) differed among populations (Fig. 4a), which reflects differences in motivation to approach purring songs, regardless of frequency (Table 1). Specifically, Kalaupapa (the population where we first discovered purring) had significantly higher responsiveness to purring songs than all other populations (Fig. 4a; Tukey's post-hoc: Kalaupapa–La'ie: $p = 0.024$; Kalaupapa–Hilo: $p = 0.017$; Kalaupapa–Wailua: $p = 0.035$). We also uncovered some population-level trends that fall short of statistical significance including variation in responsiveness when responses were measured as distance traveled, and variation in peak preference when responses were measured as positive phonotactic behavior and speaker contact (Table 1).

**Exemplar experiment: Cricket preferences for purring variants.** When we measured individual cricket responses to purring variants that differed across all song characteristics (more than just frequency), we found a similar pattern; fitness landscapes were largely flat (Fig. 5). There were no statistically significant effects of PC1 or PC2 (neither linear nor quadratic terms) on any measure of female response (phonotaxis, contact, and distance; all $p > 0.05$, Supplementary Table 3). Between 17% and 21% of crickets

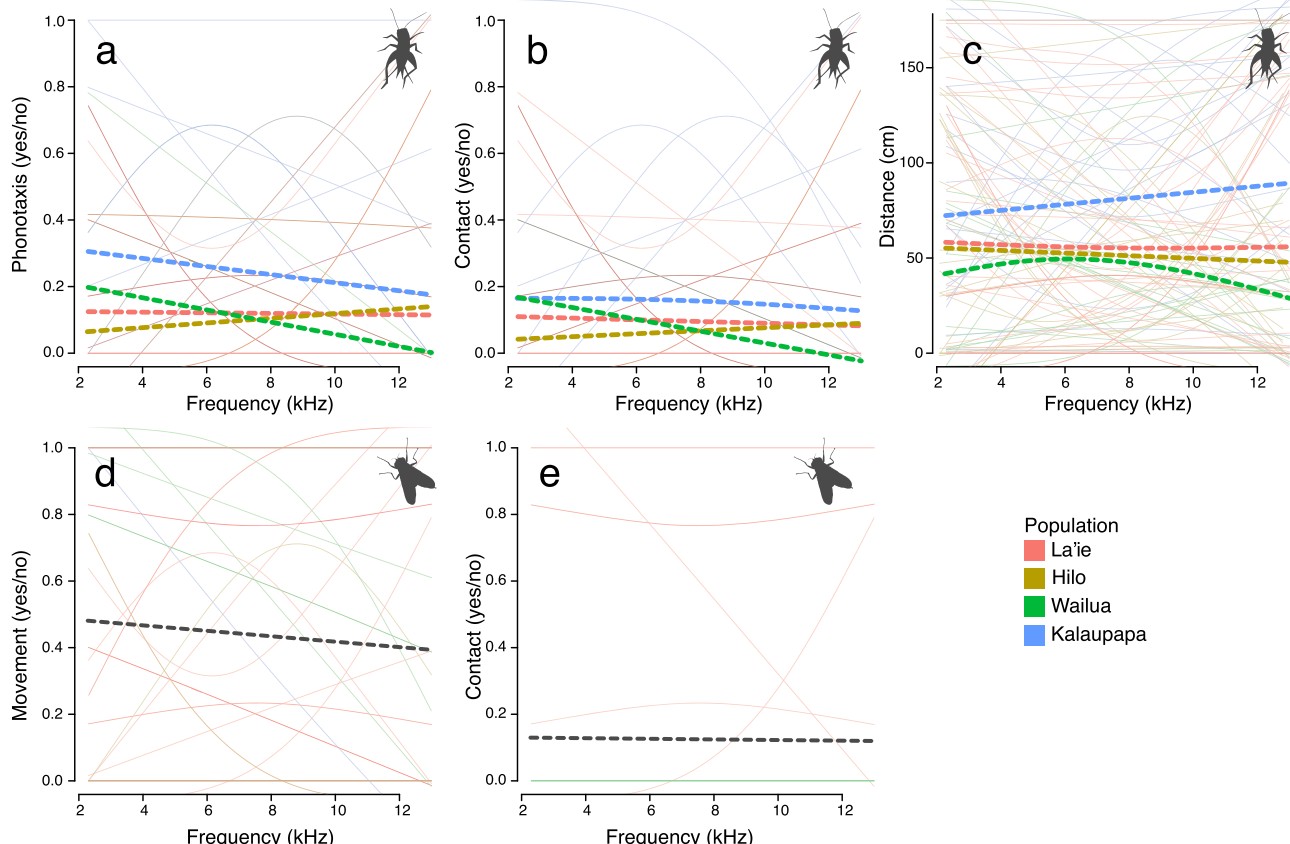

**Fig. 4 Preference functions generated from behavioral responses of female _T. oceanicus_ and _O. ochracea_ to frequency-manipulated purring songs.** Panels **a**–**c** show whether _T. oceanicus_ females ($N = 114$) displayed phonotactic behavior (**a**), whether or not females contacted the speaker (**b**), and distance traveled (**c**). Panels **d** and **e** show whether or not female _O. ochracea_ ($N = 37$) moved during stimulus playback (**d**) and whether they contacted the speaker (**e**). Thin lines show individual-level splines fit by the program Pfunc[56] to female responses (as identical splines are possible with binomial responses, some replicates overlap in panels **a**, **b**, **d**, **e**). Dashed lines show splines summarizing responses at the population-level (crickets; **a**–**c**) or species-level (flies; **d** and **e**). Source data are provided as a Source Data file.

**Table 1 Population differences in female cricket responsiveness and peak preference to frequency-manipulated purring songs. Responsiveness and peak preference parameters were extracted from individual-level preference functions ($N = 114$ female crickets) fit to three different measures of female responses to purring song frequency: distance traveled, phonotaxis (yes/no), contact (yes/no). Population differences in these parameters were tested using one-way ANOVA.**

| Model response | df | _F_ value | _P_ value |
|---|---|---|---|
| _Distance traveled_ | | | |
| Responsiveness | 3,110 | 2.68 | 0.051 |
| Peak Preference | 3,89 | 0.23 | 0.877 |
| _Phonotaxis_ | | | |
| Responsiveness | 3,110 | 4.06 | 0.009 |
| Peak preference | 3,52 | 2.51 | 0.067 |
| _Contact_ | | | |
| Responsiveness | 3,110 | 1.72 | 0.168 |
| Peak preference | 3,41 | 2.69 | 0.059 |

responded positively to each purring exemplar, supporting the contention that fitness surfaces are flat at time zero. We did find a trend towards increased positive phonotaxis with increasing values of PC1 (Fig. 5a; Supplementary Table 3). Larger values of PC1 correspond to purring songs with greater relative amplitudes in low-frequency ranges (including the ancestral peak frequency range of 3.5–6 kHz), and lower peak frequency (Fig. 2). Similar to findings from the frequency manipulation experiment, we again found much larger variances in random-intercepts and stronger intra-class correlations at the individual-level than the population-level (Supplementary Table 3), such that any two observations within an individual are much more likely to be correlated than any two observations within a population. Individual models fit with seven uncorrelated song characteristics (instead of coordinates of composite PCA variables) confirmed the finding that preference functions were flat; these models showed no significant effect of any trait on any measure of cricket preference ($0.37 < p < 0.99$).

## Discussion

We capitalized on the recent evolution of a novel song in Hawaiian populations of the field cricket _T. oceanicus_ to characterize how sexual and natural selection act on new signals at their origin. When we compared the responses of intended and unintended receivers to ancestral song, purring song, and white noise, parasitoid flies were disproportionately attracted to the ancestral signal (Fig. 3). Nearly one in five female crickets, however, responded positively to the purring song, suggesting that purring may be a nearly private mode of communication amongst crickets. Upon closer inspection of cricket and fly responses to variation within purring song (through the frequency manipulation and exemplar

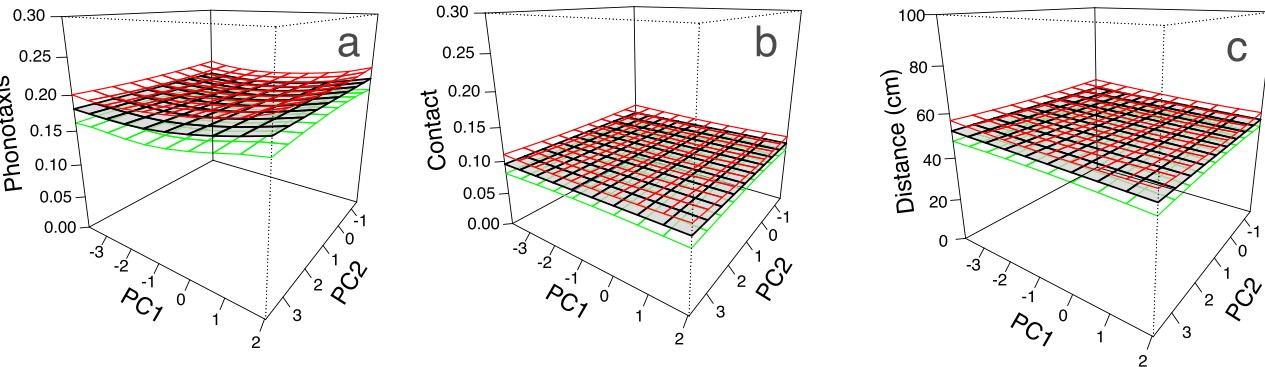

**Fig. 5 Fitness landscapes describing initial selection exerted by female crickets ($N = 271$) across the acoustic space of purring songs (exemplars) soon after the origin of purring.** Panel **a** shows whether the female cricket displayed phonotactic behavior, while **b** and **c** show whether or not females contacted the speaker and the distance they traveled, respectively. Surfaces depict female responses aggregated across populations (see Supplementary Table 3 for estimates of population random effects from GLMMs). Source data are provided as a Source Data file.

experiments), we found that aggregate preference functions are largely flat with some subtle variation among populations for crickets (Figs. 4, 5; Table 1, Supplementary Tables 1, 3). Surprisingly, we found no evidence that receivers preferred ancestral frequencies that align with hearing abilities (sensory bias) or extreme purring variants (novelty). However, flat aggregate preference functions appear to be driven by extreme among-individual variation in responses to purring song (Figs. 4, 5; Supplementary Tables 1, 3) rather than a consistent lack of preference at the individual level.

When we broadly consider how crickets and flies respond to songs produced by different male morphs (Fig. 3), we find that female crickets can and do use purring song to locate simulated mates (consistent with our previous work[41]), which would facilitate the spread of the purring morph if locating a mate translates to actual reproduction. We also find that flies exert strong selection against the ancestral morph, but not against purring morphs; in triangular arrays of speakers playing ancestral song, purring song, and white-noise, we caught 47 flies at ancestral speakers, but only caught a single fly at a purring speaker (Fig. 3b). In other words, purring song does appear capable of attracting flies using natural search behaviors in the field, but at extremely low rates. Importantly, these observed rates of attraction of flies to purring songs may even be inflated, because the amplitude at which we broadcast purring songs is at the high end of recently recorded purrs. Our current data suggest that purring is a private mode of communication amongst crickets. If it confers higher fitness than ancestral or silent types, purring may persist and increase in frequency.

Given *O. ochracea*'s poor ability to locate purring song in the field, how then do we explain their presence in populations like Kalaupapa that lack the ancestral male morph? *O. ochracea* hearing ability peaks around 5 kHz[57,58], though the flies are capable of detecting cricket songs that vary somewhat in frequency and a lot in temporal patterning[59]. It is possible that shifts (evolved or plastic) in hunting strategies, shifts in sensory abilities, and/or host song learning by *O. ochracea* may enhance the likelihood of encountering non-ancestral type hosts[60]. *T. oceanicus* developing in environments with less ancestral song, for instance, are more ambulatory and show enhanced phonotactic responses that facilitate mate location[61,62], and similar processes may occur in *O. ochracea*. Another possibility is that alternative hosts exist in these locations. *O. ochracea* parasitize at least 18 species of field crickets that sing in a similar frequency range[59], and we have (rarely) collected both *Gryllodes sigillatus* and *Gryllus bimaculatus* in Kalaupapa. Future work across populations will definitively test for neurophysiological responses to purrs by crickets and flies,

examine the role of plasticity in shaping phonotactic responses to purring song, and explore the possibility of alternative hosts and hunting strategies for the fly in Hawaii.

Our second goal was to determine whether intended and unintended receivers exhibit preferences for specific purring variants at time zero. We hypothesized that current responses to purring may depend strongly on the evolutionary history (and historical peculiarities) of the organism, generating initial preference functions with peaks at ancestral (sensory bias) or extreme (preference for novelty) song characteristics, for instance. Instead, we found flat aggregate population-level (cricket) and species-level (fly) preference functions in both the frequency manipulation and the exemplar experiments (Figs. 4, 5; Supplementary Tables 1–3). Though both female flies and crickets did prefer ancestral songs to purring songs generally (Fig. 3), they did not disproportionately respond to purring variants with more ancestral characteristics that coincide with cricket and fly hearing ability (i.e., sensory bias for peak frequency near 4.8 kHz; Fig. 4). The nearly equivalent responses of female crickets and female flies to each variant of purring we presented (Figs. 4 and 5) supports the idea that organisms perceive and can respond to a wide variety of cues and signals[15,16], and we contend that this broad capacity for response may be especially important in the origins of new sexual signals. The appearance of novel signals like purring may be ephemeral blips that are not detectable in phylogenetic reconstructions. This makes longitudinal studies that can capture novel signal and initial receiver responses in real time particularly valuable because they provide insight into microevolutionary processes immediately upon the origin of novelty.

The flat or nearly flat aggregate preference functions we uncovered are a consequence of incredible among-individual variation in female cricket and fly responses to purring song (Fig. 4, individual-level ICCs in Supplementary Tables 1–3). In each of our analyses, we uncovered substantial individual-level variation in the shape (individual splines in Fig. 4) and/or height of individual preference functions (Fig. 4, individual-level random effects in all mixed models, Supplementary Tables 1–3). This phenotypic variation in preference functions along with the large inter-individual variation in purring songs (Fig. 2)[41] aligns with models of sexual signal divergence, which require standing genetic variation in both signals and receiver preferences and choosiness at the earliest stages of trait evolution[63]. Variation in individual preference functions may stem from differences in sensory abilities of individual receivers (reviewed in ref. [64]), plastic responses to differential juvenile and adult experience (e.g. in *T. oceanicus*[61,65]; reviewed in ref. [66]), or a peculiarity of this system like recent relaxation of selection on mating-related

morphology and behavior (e.g. ref. [40]) allowing accumulation of neutral variation[67]. Rare phenotypes (like particularly responsive individuals) may be the most important drivers of the initial evolution of a new signal[68,69], especially in small island populations where drift and selection can rapidly change the make-up of populations. It is important to note that some of the individual variation that we found likely reflects differences in general responsiveness/receptivity of individual females to acoustic stimuli, rather than only differences in female discrimination among potential mates (preference). There are numerous aspects of mate choice behavior[53,70] and other sources of variation in mate choice beyond preference (e.g. choosiness[63]; responsiveness[54]) that could facilitate contact with purring males in natural contexts, influencing evolution of both the purring signal and preferences for it.

We designed these experiments to measure selection in the field in a robust way at time zero. This approach, like any, has inherent limitations that we will address in ongoing and future work. First, since we primarily captured flies at speakers broadcasting ancestral song, our laboratory phonotaxis experiments may have excluded flies that dislike the ancestral call (i.e., there is a potential selection bias). Second, measuring behavioral phonotactic responses does not allow us to distinguish between positive receiver responses that stem from signal detectability versus aspects of preference. We can address this distinction using controlled laboratory electrophysiology experiments coupled with phonotaxis trials (e.g. ref. [58]). Third, the complex genetic and plastic underpinnings of the receiver responses we uncovered in this paper remain poorly understood. Challenges include the fact that even ancestrally, females do not have sound producing structures on the wings (meaning that female genotypes at loci associated with wing morphology are unknown), male signals and female preference may be genetically linked[71], song likely covaries with other courtship traits (e.g. cuticular hydrocarbons[72]; substrate-borne vibrations[73]), and we have only just begun work to explore the role of plasticity in shaping purring signals and responses of crickets and flies. Fourth, more work is needed to reveal how female cricket and fly responses to purring song translates to variation in fitness in natural contexts. Reproductive success in the wild occurs within a complex network of potential mates and natural enemies. Behavioral trials that measure cricket phonotaxis in the very fields where these animals mate, as well as experiments manipulating the distribution of morphs of whole populations while quantifying male–male competition, mate choice, and copulations will shed further light on how sexual selection operates on the different morphs.

We capitalized on a recent evolutionary event to immediately document receiver responses to a highly variable new signal (Fig. 2)[41]. Variation in purring signals and receiver responses among individuals may exist across our study populations because (a) they are functionally neutral (no cost), (b) different purring songs are equally (un)detectable to intended and unintended receivers and therefore confer similar net fitness advantages, or (c) purring songs and accompanying responses have not been present for long enough to be shaped by selection. Our current results support the hypothesis that different purring songs do confer some advantage to signallers. We find that at least some female crickets are willing to respond to each variant of purring song we presented (17–21% positive responses across all playbacks), while natural enemies are largely unresponsive to the novel signal. If purring persists, as selection acts on signalers and receivers, we may see a reduction in inter-individual variation and selection favoring intended and unintended receivers that detect and locate purring crickets. Our planned long-term, repeated

monitoring of variation in signals and receiver responses across replicate Hawaiian populations that contain purring males will shed light on how ecology, social environment, and drift shape new signals, providing a rare opportunity to empirically assess prevailing theory about divergence by sexual selection and coevolution between signalers and receivers.

## Methods

**Field sampling**. We conducted field and phonotaxis experiments at field stations on each island (Fig. 1) between June 2018 and January 2020. Upon collection, we housed female and male crickets in same-sex 15 L storage containers at ambient temperatures with natural lighting, moistened cotton for water, egg carton for shelter, and ad libitum rabbit food for at least 24 h before testing. After collecting parasitoid flies, we transferred trapped female flies to individual $40 \times 40 \times 61$ cm mesh insect cages and held them at ambient temperatures with natural lighting, moistened cotton and local vegetation for 24 h before the trials described below. We complied with all relevant ethical regulations for animal testing and research throughout the experiments. Our study organisms are exempt from IACUC approval.

**Signal recording**. Quality, standardized recordings of purring and ancestral songs were instrumental to producing the frequency-manipulated and exemplar songs we played to receivers in the following experiments. We recorded long distance calling songs of first-generation, lab-reared male crickets from Kalaupapa, La'ie, and Manoa in an acoustically isolated and temperature-controlled recording studio at the University of Denver. We isolated adult males for at least 5 days prior to recording to encourage them to produce long-distance calls. In the recording studio, we placed each male in an open-topped 0.5 L deli cup positioned 40 cm from a Sennheiser MKH800 microphone (Sennheiser, Wedemark, Germany) set to omnidirectional detection. We used a Millennia HV-3D preamplifier (Millennia, Diamond Springs, California) with gain set to 48 dB and recorded inputs through an Avid HD analog to digital converter (Avid, Burlington, Massachusetts) with sampling rate of 192 kHz at 24 bit depth. We processed .wav files in Audacity (http://audacityteam.org) and removed ambient background noise using the Noise Reduction function.

**Search behavior in response to ancestral song, purring song, and white noise**. Our first question was how female crickets and flies respond to ancestral songs, purring songs, and negative controls under conditions that mimic their natural search behaviors in the field. To answer this for female crickets, we analyzed responses to the ancestral song, unmanipulated purring song, and white noise tracks, compiling these subset responses from the phonotaxis experiments (frequency manipulation experiment and exemplar experiment) that are fully described below. To determine whether flies are attracted to purring songs from long distances under natural conditions, and how their attraction to purring songs compares to ancestral song and white noise, we conducted fly choice tests in the field in all six populations in June 2019 and again December 2019–January 2020. In each replicate (La'ie = 30; Hilo = 16; Wailua = 16; Manoa = 4; Kalaupapa = 10; Kapa'a = 4), we placed three funnel-shaped sound traps (following ref. [57]) fashioned from 2 L plastic bottles in an equilateral triangle 10 m apart from one another and broadcasted the unmanipulated purring track, the ancestral track, and white noise simultaneously from the three traps (location of each determined using a random number generator). We broadcast stimuli at biologically realistic volumes (70 dBA at 1 m away for the typical song, 53 dBA at 1 m away for purring and white noise) from AGPTEK A02 MP3 players for approximately two hours each evening when flies are most active (beginning approximately one hour before sunset). At the end of the playback, we counted the number of flies in each funnel trap and released the flies back into the fields in which we caught them.

**Frequency manipulations: Cricket and fly responses to purring song peak frequency**. Though the peak frequency of songs varies among cricket species, it tends to be relatively invariable within species (e.g. ref. [74]; ancestral calling song in ref. [41]) making it useful in conspecific identification[46–48]; however, purring song peak frequencies vary dramatically among males[41]. Thus, we first varied the peak frequency of purring songs used in phonotaxis trials. In preliminary exploration of song variation, we found that although purring songs are quite broadband (covering a wide range of frequencies[41]), there are several frequencies within most of the recorded songs where there are minor spikes in acoustic power. We used the Plot Spectrum feature in Audacity (version 2.3.1 http://audacityteam.org) to identify these frequency spikes, which occur at approximately 2.3, 4.75, 7.5, 10, and 13 kHz. We produced five frequency-manipulated song tracks from an identical track of looped calling songs belonging to four randomly selected males (as in ref. [41]) in Logic Pro X (version 10.4.8) by reducing the amplitude of all frequencies in the song by 10 dB, and then boosting the amplitude of the focal frequency by 10 dB using the Channel EQ plugin (version 10.4.6, Apple Inc., settings: master gain = $-10$ dB, band gain = $+10$ db, Q = 1.20) (Supplementary Fig. 1). All

manipulated and unmanipulated purring songs were then set to the same overall amplitude (53 dBA) before playback.

In June 2018 and December 2018–January 2019 we collected wild adult female *T. oceanicus* and *O. ochracea* (using the funnel traps described above) in Kalaupapa, La'ie, Hilo, and Wailua. Then, in standardized phonotaxis trials, we tested each female cricket with eight stimuli: the five frequency-manipulated purring tracks, an unmanipulated purring track, a negative control (white noise), and a positive control (ancestral song), played back in random order, except that the ancestral song was played last. The phonotaxis arena was 50 cm × 195 cm × 25 cm, with an ECOXGEAR EcoXBT portable waterproof bluetooth speaker broadcasting stimuli at biologically realistic volumes (70 dBA at 1 m away for the typical song, 53 dBA at 1 m away for all purring variants and white noise), positioned 175 cm from the end of the arena where the female was released. The playback of each stimulus lasted a maximum of one minute or until the female contacted the speaker. Females who exhibited positive phonotaxis were given an additional minute to make contact with the speaker (or not). Between trials we wiped down the arena using a bleach solution. We recorded whether or not the female exhibited positive phonotaxis (regardless of whether she reached the speaker in the allotted time), whether she contacted the speaker, and distance traveled toward the speaker (positive phonotaxis and contacting the speaker indicate greater preference for a particular song variant). We considered females to be phonotactic when they generally meandered (moved in a zig–zag pattern[75,76]) in the direction of the audio stimulus without simply following the wall and without circling indefinitely, and 2–3 trained observers had to agree upon this decision independently for each playback. The three dependent variables (phonotaxis, contact, and distance) captured different elements of female responses, but were correlated (Phonotaxis × Contact = 0.75, Distance traveled × Phonotaxis = 0.54, Contact × Distance traveled = 0.49). Females were only removed from analyses if they never responded to any playback of any song variant.

Each female fly was also tested with all eight stimuli at the same amplitudes reported in the fly field experiment at dusk during the flies' most active phase. Each phonotaxis test consisted of gently jostling the fly until she moved to the top of the mesh cage and then broadcasting each song stimulus in turn from an ECOXGEAR EcoXBT speaker positioned in one of the four bottom corners of the mesh cage (corner and song stimulus order were determined using a random number generator). We measured whether the female moved during playback and whether or not the female contacted the speaker.

**Exemplar experiment: Cricket responses to purring variants**. One year later, after recording purring males from additional populations, we chose eight representative songs (hereafter, exemplars) that represent the many dimensions of variation in purring song. We first extracted 11 song characteristics (Table 2) from the first complete song in the first continuous bout of calling from recordings of 46 purring males (6 La'ie, 16 Manoa, and 24 Moloka'i). To facilitate comparison of songs across the multidimensional data set, and to account for significant correlations between song characteristics (Fig. 2, Supplementary Fig. 2), we performed a principal components analysis on z-scores of all song characteristics and visualized songs across the first two PCA axes, which captured a large amount of acoustic variation (51%, Fig. 2). PC1 largely captured frequency measures, while two measures of the distribution of acoustic power across frequencies, broadbandedness and number of peaks, loaded strongly onto PC2 (Fig. 2). We chose eight exemplars that span the full range of variation in purring song, selecting several songs that represent phenotypic extremes and including exemplars from each of the populations from which we had purring recordings (Fig. 2; exemplars span the 2–93% and 17–100% percentiles of PC1 and PC2 scores, respectively).

In June 2019 and December 2019–January 2020 we collected wild female *T. oceanicus* in Kalaupapa, La'ie, Manoa, Hilo, Wailua, and Kapa'a for the exemplar experiment. Crickets were held at field stations as described above and tested using the same phonotaxis arena as in the frequency manipulation experiment, with two changes. First, while the dimensions of the arena were the same, we placed the speaker one meter from the female cricket, reducing the travel distance to match previously published work[41]. Second, we replaced the frequency-manipulated songs and the unmanipulated purr with the set of eight exemplar songs. All eight exemplars and the white noise control were played in a random order at a

biologically realistic amplitude of 53 dBA. Two final songs were played at a higher amplitude (70 dBA) and in the same order so as not to bias female behavior: first, one exemplar was replayed a second time, but at the same amplitude as an ancestral song, and last, the ancestral calling song loop was played at its realistic amplitude. Female responses were measured as in the frequency manipulation experiment (correlations of response variables: Phonotaxis × Contact = 0.70, Distance traveled × Phonotaxis = 0.51, Contact × Distance traveled = 0.44).

**Statistics**. To test for differences in responses of female crickets to purring, ancestral and white noise (our negative control) we ran a generalized linear mixed model with binomial errors and random intercepts in the R (R Core Team, 2020) package lme4[77], with phonotaxis (yes/no) as the response variable, song type (ancestral/purr/white noise) as an independent variable and random effect structure of individual nested within population.

To visualize female cricket and fly responses, we first constructed univariate individual-level and population-level preference functions in PFunc[56]. PFunc fits nonparametric curves (cubic splines) through data on receiver responses to continuous variation in sexual signals in order to generate mate preference functions, which limits the extent to which the researcher imposes shape on the data (e.g. refs. [78,79]). These preference functions capture aspects of receiver responses to variation in mating traits, including peak preference, peak height, tolerance, responsiveness, and preference strength (defined in ref. [56]). For each measure of female cricket or fly response, we generated functions representing individual's responses to the five frequency exemplars.

To determine whether female crickets prefer certain purring song frequencies over others (frequency manipulation experiment), we next constructed random-intercept mixed effects models in the R package lme4. We ran separate models for each measure of female response (LMM for distance; GLMM with binomial error for phonotaxis and contact), using standardized (z-scored) frequency and frequency² terms (mating preferences are often nonlinear in response to stimuli) as fixed effects, and population and individual as nested random effects. The coefficients for the linear and quadratic frequency terms in the LMMs and GLMMs reveal whether or not the frequency variants played predict female responses. The random effects on individual and population in these models allow us to compare relative variance in responses among individuals and among populations. We also calculated intra-class correlations of random effects in the package performance[80] to estimate the proportion of variance explained by each grouping variable in the models (population and individual), and approximated p-values of random effects using likelihood ratio tests. We calculated p-values of fixed effects in LMMs and GLMMs using Satterthwaite's method in the package lmerTest[81], and Type II Wald Chi Square tests, respectively. We used the same approach to test whether flies respond more strongly to certain purring song frequencies over others, except that we removed population from the model because small sample sizes for some populations prevented models from converging. Finally, we extracted two uncorrelated preference function traits from the splines generated in PFunc (peak preference (the most preferred song variant) and responsiveness (the height of the function)), and used one-way ANOVAs with population as a predictor to determine whether populations differed in these parameters.

Songs used in the exemplar experiment differed in multiple song characteristics (Table 2), necessitating a different modeling approach for assessing cricket responses. For each response variable (distance, phonotaxis, contact), we fit complete second-order mixed effects models with the coordinates of exemplar songs along the first two PCA axes (Fig. 2) as our two predictor variables (PC1, PC2). Models allowed for random intercepts for individual nested within population, and we calculated p-values of fixed effects with Type III Wald Chi Square tests. To visualize preference surfaces in two-dimensional space, we plotted thin-plate splines in mgcv[82], using smoothing terms for PC1 and PC2, and random effects for individual and population. Because using model outputs (e.g. PCA coordinates) as inputs in subsequent analyses can introduce error, we refit models of each response variable using a subset of raw song characteristics (Table 2) in which no two variables had correlation coefficients >0.5 (using the findCorrelation function in the caret R package[83]; uncorrelated variables retained: peak frequency, peak frequency bandwidth, and relative amplitudes of the 2–3.5, 3.5–6, 9.5–12.5, 12.5–17.5, 17.5–20 kHz frequency ranges).

**Table 2 Song characteristics included in a principal components analysis used to choose purring exemplars.**

| Song characteristic | Description |
| --- | --- |
| Peak frequency | Frequency with the greatest acoustic power. |
| Peak frequency bandwidth | Difference between upper and lower frequencies 10 dB below the peak frequency band's apex. |
| Number of peaks (frequency bands) | Number of frequency bands between 1.5 and 20 kHz that fall within 10 dB of the peak frequency band's apex. |
| Broadbandedness | Range (in Hz) of frequency bands 10 dB below the peak frequency band's apex. |
| Relative amplitude of 6 frequency ranges | dB difference between each defined frequency range and the song's total amplitude. Frequency ranges include: 2–3.5, 3.5–6, 6–9.5, 9.5–12.5, 12.5–17.5, 17.5–20 kHz. Ranges determined based on cricket hearing ability[50]. |
| Proportion long chirp | Length of the long chirp (in ms) divided by the sum of the long and short chirp lengths. |

**Reporting summary**. Further information on research design is available in the Nature Research Reporting Summary linked to this article.

## Data availability
The authors declare that the data supporting the findings in this study are available within this paper and in Supplementary Data 1. Source data are provided with this paper.

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

## Acknowledgements

We would like to thank Kalaupapa National Historical Park and the residents of the community for supporting our fieldwork, especially our research coordinator Paul Hosten and other local facilitators including Jeanine Rossa, Pastor Richard Miller, and Barbara Jean Wajda. Thank you to the Lamont School of Music for access to recording studio space and to our sound engineers Dr. Michael Schulze, Kyle Hughes, and Colton Sparks. Thank you to all of the DU students who assisted with cricket husbandry, recording, and field work especially Makenzie Day, Sophia Anner, Jake Wilson, Claudia Hallagan, and Brooke Washburn. Cathy Durso provided instrumental feedback and advice on statistical analysis. This work was supported by research grants from the National Science Foundation (IOS 1846520) and the American Philosophical Society to RMT, the Orthopterists' Society to EDB, Brooke Washburn, and Jake Wilson, the Stoffel Fund for Excellence in Scientific Inquiry Grant to E.D.B., Sigma Xi and Moras and Erne Shubert Graduate Fellowship Fund grants to J.H.G., and donations from Kalaupapa National Historical Site (lodging) and generous individual donors (Thomas and Anne Dale Blair, Lyndy Broder, Jennifer Priester, H.T. Malasadas, and Frank Truslow) to Kickstarters "Purring Cricket Discovery" and "Evolving Crickets that Purr."

## Author contributions

All authors collected data and wrote the manuscript. R.M.T. and E.D.B. conceived of the ideas, designed the experiment, and wrote the first draft of the manuscript. D.M.Z. performed statistics and made figures. J.H.G. and A.W.W. analyzed song characteristics to determine exemplars. J.H.G. generated frequency manipulations, song exemplars, and spectrogram figures.

## Competing interests

The authors declare no competing interests.
