## [Peer Review File · Nature Communications]

Reviewers' Comments:

Reviewer #1:

Remarks to the Author:

Extreme variation in responses of intended and unintended receivers to a new sexual signal

This novel manuscript investigates the earliest steps in the evolution of novel sexual signals. Mutations can result in inadvertent cues. These novel cues can, in turn, be selected through coevolution between signalers and receivers to become sexual signals. However, extremely little is known about how receivers produce a coupled response to these novel signals. The authors investigated this elusive process using the Pacific field cricket and a novel mutation that produced purring male crickets. "Purring" has recently been observed in some male crickets across the Hawaiian Islands. The authors quantified the natural and sexual landscape that shape these novel traits. Specifically, they quantified preference functions and the resulting fitness advantages across multiple islands, testing alternative hypotheses about how selection may act on these novel signals very early in their evolutionary origins. While female field crickets responded favourably to these novel male cues, unintended receivers (parasitoid flies) did not, revealing that purring may be an evolutionary solution to extreme parasitism. As a result, male purring appears to be an 'evolutionary solution' to conflicting selection pressures and act as a private communication channel between crickets (intended receivers) that is not easily eavesdropped on by unintended receiver parasitoid flies.

The introduction nicely describes the importance of the core question (evolution of novel sexual signals), the field cricket and parasitoid fly system on the Hawaiian Islands, the evolution of silent crickets because of the extreme selection by parasitoid flies, the putative mutation of purring in males across several of the islands, and then sets up the multiple hypotheses that the authors tested about the selective landscapes that can favour novel signal establishment. Specifically, the authors tested the sensory bias hypothesis (females prefer songs with peak frequencies that match ancestral song), the novel stimuli hypothesis (females prefer rare song variants), and the lagging hypothesis (females exhibit flat preference functions as they lag behind the origin of purring).

The methodology is well laid out and appropriate for trapping female parasitoid flies using sound (originally described by Tom Walker and using a nicely randomized setup) and for quantifying female cricket responses to the different songs. A mention should be made here towards Tom Walker's work and methods, given he was the first to describe trapping *Ormia* in this way. I was pleased the authors used the techniques described by PFunc for visualizing female preference functions by Fowler-Finn, Rodriguez and Kilmer's papers, as their approach is one of the strongest for this kind of work. Throughout the approach of randomizing sound production out of the different speakers (parasitoid fly attraction in the field) and which songs are played when for the female cricket and parasitoid preference functions shows careful attention to detail, attention that is important for this type of work. Given the descriptions, my lab would easily be able to reproduce the work, so the methods seem quite appropriate.

The statistics (GLMM; LMM; ANOVA) for quantifying phonotactic behaviour (presence/absence; distance travelled; speaker contact; peak performance; responsiveness) appear appropriate (including the use of random effects for individual and population level for the GLMM and LMM's) for everything but the exemplar experiment. The stats for quantifying I have less confidence in the statistics associated with the exemplar experiment, largely because the authors are running statistics on statistics. While this is problematic, I see no other way around it, other than to raise that this may be a problematic approach. The problem here is that the exemplar songs differed in so many ways (11 different traits measured) that there is really no way to handle this other than running statistics on the first two PCA predictor variables.

The results are exceptionally clear. While the manuscript does a fantastic job of showing that the flat overall preference functions are driven by high inter-individual variation and not by a consistent lack

of preference, I was initially concerned that this may be due to high intra-individual variation (i.e., low repeatability). I was therefore very pleased to see that the supplemental Table S1 dealt with exactly that issue, with the authors describing repeatability of preference functions, showing that the intra-class correlations between observations of the same individuals are much stronger than they are between different individuals.

Overall, these results reveal that while female crickets respond positively to purring songs, parasitoid flies do not, suggesting that purring is a 'private' mode of communicating between crickets and excluding parasitoid flies. Aggregate preference functions are flat but vary subtly across island populations. These results suggest no support for the sensory bias hypothesis or the preference for novelty hypothesis. The discussion of purring and what may have occurred first (purring or female preference, especially on Kalaupapa, was great, especially surrounding the extreme among-individual variation in female cricket and parasitoid fly responses. This too was very timely, especially considering Dochterman and colleague's calls for focusing on individual variation instead of means. I also agree with the authors that this variation aligns with models of sexual signal divergence.

The discussion of *Ormia*'s seemingly poor ability to locate purring song in the field (line 438) would be augmented by mention of *Ormia*'s hearing range (e.g., some of the older work coming out of Ron Hoy's lab) and the likelihood that they can even hear these purring cricket songs.

The problems outlined in lines 522 – 540 are appropriate. This would be the place to add in a few lines about the problems associated with running statistics on statistics.

Overall, the authors have run a superb and fascinating set of experiments that capitalize on a 'unique evolutionary event'. Together, these experiments and their fascinating results suggest that different purring songs confer an advantage to the purrers. This seemingly alternative mating strategy has the added benefit of avoiding parasitoid attraction. It will be fascinating to determine whether selection erodes this extreme variation that the authors have uncovered. I thoroughly enjoyed reading this manuscript as it offers plenty of food for thought about the evolution of signalers and receivers following novel mutations, especially in a complex selective environment. Given this, I feel it will be highly relevant for anyone interested in behaviour, signalers and receivers and evolution. I anticipate that purring and this story will become part of the classic textbook examples that we all talk about when teaching about the evolution of signaler receiver systems.

Best of luck with revisions and getting this very fine work published.

Sincerely,

Susan M. Bertram

Reviewer #2:

Remarks to the Author:

This is a very interesting study of an important evolutionary process in a very interesting study system. It should be of interest to a broad audience. The questions are clearly posed, the methods are sound, and the conclusions are very sensible. I do have a few minor questions-suggestions.

In the title, disambiguate the meaning of "variation in the responses" by clarifying the distinction between variation within and variation between populations.

On line 20, the meaning of "evolutionary solution" needs clarification – is this an ESS? Possibly, but it is probably only a matter of time and chance until the private channel is no longer private. Many readers could use some help understanding the meaning of an "Evolutionarily Stable Solution."

The issue of the stability of the purring "solution" needs more discussion. Line 117 says "the characteristics of purring songs are incredibly variable among individuals." The pattern in Figure 1 is consistent with the hypothesis that the purr is an unstable transitional state between ancestral and silent – that the data are catching snapshots of an unstable transitional state that is being replicated in several locations and populations. This should be discussed as a possibility when setting up line 562: "Our planned long-term, repeated monitoring of variation in signals and preferences across replicate Hawaiian populations ...".

Line 148 says "If crickets respond positively to purring songs, but flies do not, this would suggest that purring is a private mode of communication amongst crickets." Yes, and it would suggest an ESS, given current genetic variation. So, this is a crucial question that needs additional consideration in the presentation. Please address it.

Line 541 says "We capitalized on a unique evolutionary event ...". Aren't the different populations meant to be replicating independent events? Reconcile this seeming inconsistency between "unique" and replication.

All of my concerns are conceptual and can easily be addressed in the presentation.

Reviewer #3:

Remarks to the Author:

This is a fascinating study system and I congratulate the authors (whoever they are) on their hard work. I am well aware that conducting so many phototaxis trials per individual is a lot of effort. And the amount of work is further compounded by conducting the same phonotaxis experiments at multiple (4-6) sites. Unfortunately, at least in its present form, I do not think the study is appropriate for Nature Communications. This is not to say that it would not be well suited to a very good journal like Am Nat or Evolution.

In overview, my reasons for saying this are:

1. The writing needs to be tightened up, especially in the Introduction.

2. I am not yet convinced that the variation in 'preferences' among individuals is due to differences in choice as opposed to activity levels at the time of capture. I know you have higher ICC for individuals than populations, but this could simply reflect that some individuals moved around (= distance + likelihood they contacted the speaker + likelihood they showed what appears to be phototaxis; I might be wrong about this last response, however, but the MS is not clear as to how you would distinguish 'positive phototaxis' from simply moving in the general direction of the speaker).

3. When it comes to the variation among individuals (Fig 4), it is not clear to me the extent to which this might simply reflect sampling error. I know you have positive values for ICC, but you do not report whether or not these differ significantly from zero. In Fig 4 it also appears that the number of females per site sampled must have been low if all individual preference functions are being drawn (except for Fig4 c). If the lack of line is because most females did not shown phonotaxis or make contact (Fig4 a and b) (i.e. most the lines are at 0) then this suggests that part of the reason for the flatness of the population mean functions is simply due to most females not being very responsive. One line of argument that might help your case would be any evidence (perhaps from already published studies) that comparable individual preference functions for variation in ancestral calls are less variable (i.e. lower ICC) than those for purr call variation.

Specific comments

Line 53. Might there be studies of flowers that provide examples?

Line 101. Please remind the reader when the flies reached Hawaiian islands.

Line 111. Perhaps the data is in Ref 59, but it would be good to explicitly state whether there is evidence that purring is a heritable trait and, like silent, whether it can be attributed to a single mutation.

Figure 1 - I would provide information for each site on:

(a) the % of each morph present, and info as to whether this is repeatable across years

(b) The density/abundance of the flies

Line 140 - add a brief explanation as to why you know the sites are evolving independently.

Line 164. I know it is possible, but it does seem implausible a priori that there is a preference for novel signals. Has this ever been shown in an insect? Even the vertebrate examples in which novelty has been shown are not especially well supported cases.

Line 202. Given you had replicates at each site, Fig 3b should include a SD or SE around the mean.

Line 227. Personally I think it is OK, but it is worth noting that by collecting females attracted to ancestral calls, you reduce the likelihood that you then tested females that 'dislike' the ancestral call or prefer purrs. i.e. there is a selection bias.

Line 308 Was it Type II or Type III ? Please confirm.

Line 361. What was the effect size? Is this "SD" in the Supp tables? Please clarify.

Fig 4 Please state the sample size (number of individuals tested) for the five figures. Giving the value for each site in each figure.

****NB**** Line 379. Table 2 reports a pop difference in responsiveness to purrs, but Fig 3 does not appear to show this. Can you please expand on the apparent difference?

Figure 5. Are the surfaces shown for multiple sites? If so, this is not apparent from the figure.

Discussion. I could be wrong, but I think that you need to tackle head on the concern that the extreme variation among individuals is due to sampling error. I guess the main argument against this would be to test that the ICC is significantly greater than zero (I might have missed it, but this is not reported). However, it is still not clear to me the extent to which the behavioural measures you used reflect mating preferences (as this is a one choice trial set up) rather than variation in a female's general propensity to move, which might vary for reasons unrelated to mating.

p26 Typo - no number for the Darwin reference.

Discussion: It is too long and can be shortened with no loss of value. Parts of it are too speculative. I take your point about non-significant effects being of potential biological significance. but one can equally (and more appropriately) agree that NS means either there is nothing happening or that you really need to boost your sample sizes to have enough power to detect smaller effects of biological interest.

Other: Please report the bivariate correlations between distance, contact and phonotaxis. To what extent are these independent measures of preference?

In sum, despite my reservations and calls for clarity, I do think this paper reports on a fascinating

phenomenon. The core result that females respond to purrs and flies do not (yet!) does make for a clear prediction that the proportion of males purring will increase significantly in the next few years. If the authors do indeed find this is the case, then they will definitely have a great paper documenting evolution in action.

I hope my comments are useful and constructive. (And apologies for my typos - it is after midnight).

Signed: Michael Jennions

REVIEWER COMMENTS

Reviewer #1 (Remarks to the Author):

Extreme variation in responses of intended and unintended receivers to a new sexual signal

This novel manuscript investigates the earliest steps in the evolution of novel sexual signals. Mutations can result in inadvertent cues. These novel cues can, in turn, be selected through coevolution between signalers and receivers to become sexual signals. However, extremely little is known about how receivers produce a coupled response to these novel signals. The authors investigated this elusive process using the Pacific field cricket and a novel mutation that produced purring male crickets. "Purring" has recently been observed in some male crickets across the Hawaiian Islands. The authors quantified the natural and sexual landscape that shape these novel traits. Specifically, they quantified preference functions and the resulting fitness advantages across multiple islands, testing alternative hypotheses about how selection may act on these novel signals very early in their evolutionary origins. While female field crickets responded favourably to these novel male cues, unintended receivers (parasitoid flies) did not, revealing that purring may be an evolutionary solution to extreme parasitism. As a result, male purring appears to be an 'evolutionary solution' to conflicting selection pressures and act as a private communication channel between crickets (intended receivers) that is not easily eavesdropped on by unintended receiver parasitoid flies.

The introduction nicely describes the importance of the core question (evolution of novel sexual signals), the field cricket and parasitoid fly system on the Hawaiian Islands, the evolution of silent crickets because of the extreme selection by parasitoid flies, the putative mutation of purring in males across several of the islands, and then sets up the multiple hypotheses that the authors tested about the selective landscapes that can favour novel signal establishment. Specifically, the authors tested the sensory bias hypothesis (females prefer songs with peak frequencies that match ancestral song), the novel stimuli hypothesis (females prefer rare song variants), and the lagging hypothesis (females exhibit flat preference functions as they lag behind the origin of purring).

The methodology is well laid out and appropriate for trapping female parasitoid flies using sound (originally described by Tom Walker and using a nicely randomized setup) and for quantifying female cricket responses to the different songs. A mention should be made here towards Tom Walker's work and methods, given he was the first to describe trapping *Ormia* in this way. I was pleased the authors used the techniques described by PFunc for visualizing female preference functions by Fowler-Finn, Rodriguez and Kilmer's papers, as their approach is one of the strongest for this kind of work. Throughout the approach of randomizing sound production out of the different speakers (parasitoid fly attraction in the field) and which songs are played when for the female cricket and parasitoid preference functions shows careful attention to detail, attention that is important for this type of work. Given the descriptions, my lab would easily be able to reproduce the work, so the methods seem quite appropriate.

Excellent. We are so glad that you found the introductory materials clear and the methods sufficient to reproduce the work. Your description here is a wonderful summary of what we did and why. Thank you for pointing out that Tom Walker initially developed the methods for trapping *Ormia*. We have added a reference to Walker (1989) (pg 7, line 173).

The statistics (GLMM; LMM; ANOVA) for quantifying phonotactic behaviour (presence/absence; distance travelled; speaker contact; peak performance; responsiveness) appear appropriate (including the use of random effects for individual and population level for the GLMM and LMM's) for everything but the exemplar experiment. The stats for quantifying I have less confidence in the statistics associated with the exemplar experiment, largely because the authors are running statistics on statistics. While this is problematic, I see no other way around it, other than to raise that this may be a problematic approach. The problem here is that the exemplar songs differed in so many ways (11 different traits measured) that there is really no way to handle this other than running statistics on the first two PCA predictor variables.

We agree that using outputs from statistical models as inputs in subsequent analyses can propagate errors in problematic ways. You are right that our reasoning for this approach was driven by having many partially correlated song variables that would have resulted in a large number of model parameters. We have attempted to address your excellent point in several ways within the manuscript. First, we have added text to the methods (pg 12, lines 293-298) to explain the potential problems with running statistics on statistics. Second, we have repeated our analyses of all three measures of cricket responses (Phonotaxis, Contact, Distance travelled) using seven of our least correlated song characteristics as predictors. We present these new analyses in addition to the original analyses which use PCA coordinates (description of how we arrived at these seven song characteristics on pg 12, line 295). These additional analyses confirmed that preference surfaces were completely flat in this experiment, as no individual song character significantly predicted female cricket responses to variation in purring exemplars (pg 15, line 364).

The results are exceptionally clear. While the manuscript does a fantastic job of showing that the flat overall preference functions are driven by high inter-individual variation and not by a consistent lack of preference, I was initially concerned that this may be due to high intra-individual variation (i.e., low repeatability). I was therefore very pleased to see that the supplemental Table S1 dealt with exactly that issue, with the authors describing repeatability of preference functions, showing that the intra-class correlations between observations of the same individuals are much stronger than they are between different individuals.

We are glad that you noticed this important table and agree with your interpretation of the intra-class correlations. In response to reviewer 3, we have also added more detail to the manuscript to hopefully help readers interpret these measures. First, we have expanded our description of how we determined whether or not females were displaying positively phonotactic behavior (pg 8, line 203). While only tangentially related to your point, we hope to allay any concerns over sampling error (which could result in the reduced repeatability that you feared) by highlighting our standardized approach. Second, we have estimated P-values for the significance of our

random effects (for both individuals and populations) in our mixed models (pg 11, line 276; supplementary table 1, 2, and 3). While we agree with your interpretation that our observed ICCs for the individual-level are quite high, we hope that the addition of more familiar P-values will help some readers come to this same conclusion.

Overall, these results reveal that while female crickets respond positively to purring songs, parasitoid flies do not, suggesting that purring is a 'private' mode of communicating between crickets and excluding parasitoid flies. Aggregate preference functions are flat but vary subtly across island populations. These results suggest no support for the sensory bias hypothesis or the preference for novelty hypothesis. The discussion of purring and what may have occurred first (purring or female preference, especially on Kalaupapa, was great, especially surrounding the extreme among-individual variation in female cricket and parasitoid fly responses. This too was very timely, especially considering Dochterman and colleague's calls for focusing on individual variation instead of means. I also agree with the authors that this variation aligns with models of sexual signal divergence.

We are so glad that you found our work to be timely.

The discussion of *Ormia*'s seemingly poor ability to locate purring song in the field (line 438) would be augmented by mention of *Ormia*'s hearing range (e.g., some of the older work coming out of Ron Hoy's lab) and the likelihood that they can even hear these purring cricket songs.

This is a great suggestion and we now summarize what is known about *Ormia*'s hearing, including references to some of Hoy and collaborators' work in this area (pg 16, line 398-400). We now say "*Ormia ochracea* hearing ability peaks around 5kHz (Robert et al. 1992, Robert et al. 1994, Mason et al. 2001) though the flies are capable of detecting cricket songs that vary in frequency and temporal patterning (Gray et al. 2019)".

The problems outlined in lines 522 – 540 are appropriate. This would be the place to add in a few lines about the problems associated with running statistics on statistics.

In the methods, we added a brief discussion of the issues of running statistics on statistics and describe new analyses to test whether or not this poses a problem (pg 12, line 293+). In the results, we show that models using multiple, individual song characteristics led to the same conclusions as our original preference surface analysis which uses composite PCA variables in the Results (pg 15, line 364). Because we addressed it earlier in the manuscript, we have opted not to include further discussion of this point in the Discussion (because we were asked to streamline our Discussion).

Overall, the authors have run a superb and fascinating set of experiments that capitalize on a 'unique evolutionary event'. Together, these experiments and their fascinating results suggest that different purring songs confer an advantage to the purrers. This seemingly alternative mating strategy has the added benefit of avoiding parasitoid attraction. It will be fascinating to determine whether selection erodes this extreme variation that the authors have uncovered. I

thoroughly enjoyed reading this manuscript as it offers plenty of food for thought about the evolution of signalers and receivers following novel mutations, especially in a complex selective environment. Given this, I feel it will be highly relevant for anyone interested in behaviour, signalers and receivers and evolution. I anticipate that purring and this story will become part of the classic textbook examples that we all talk about when teaching about the evolution of signaler receiver systems.

Best of luck with revisions and getting this very fine work published.

Sincerely,

Susan M. Bertram

Thank you so much for your positive feedback on our manuscript and for your thorough review. We have made efforts to address all of your outlined concerns and hope you will find the manuscript much improved. We also hope that, like you, others in the fields of behavioral ecology and evolutionary biology will find this work to be of broad interest and importance.

Reviewer #2 (Remarks to the Author):

This is a very interesting study of an important evolutionary process in a very interesting study system. It should be of interest to a broad audience. The questions are clearly posed, the methods are sound, and the conclusions are very sensible. I do have a few minor questions-suggestions.

In the title, disambiguate the meaning of “variation in the responses” by clarifying the distinction between variation within and variation between populations.

This is an important distinction and we can see how the title may have been confusing before. We made some changes to the title so that it more fully encompasses the breadth of questions addressed in the paper. In response to comments from reviewer 3, much of our discussion of differences among populations is no longer a part of the paper, so it seemed appropriate to no longer include this in the title. The new title is “Singing in secret: Initial responses of intended and unintended receivers to a newly evolved sexual signal”, which we hope draws attention to the key finding that purring may protect the crickets from natural enemies.

On line 20, the meaning of "evolutionary solution" needs clarification – is this an ESS? Possibly, but it is probably only a matter of time and chance until the private channel is no longer private. Many readers could use some help understanding the meaning of an “Evolutionarily Stable Solution.”

Thank you for pointing out that this was less than clear. We have changed our language--we did not intend for the phrase “evolutionary solution” to necessarily suggest that purring is an

evolutionarily stable strategy (ESS). We do not feel quite comfortable yet claiming that purring is an ESS, especially since we have not measured lifetime reproductive success of different morphs and because this paper intentionally captures the dynamics immediately following the evolution of a new trait (initial variation). We agree that if purring confers higher fitness than the other morphs present in these populations (because it is a private mode of communication) that it may indeed be an ESS. We now say this explicitly on pg 16, lines 394-396. Additionally, we take your point below that we may be observing snapshots of a system that is not stable at the moment. We agree that it could very well be just a matter of time until the private mode of communication is breached by flies, and morph frequencies are clearly in flux in our replicate populations (see response to reviewer 3). This is all very exciting to us!

In direct response to your point here, we clarified our text in the abstract, which now states, “In field studies, female crickets responded positively to purrs, but parasitoid flies did not, suggesting purring may be a new evolutionary solution to conflicting selection pressures (a novel fitness peak), allowing private communication among crickets” (pg 1, lines 18-21). See also our responses below which highlight other parts of the manuscript where we clarified these points.

The issue of the stability of the purring “solution” needs more discussion. Line 117 says “the characteristics of purring songs are incredibly variable among individuals.” The pattern in Figure 1 is consistent with the hypothesis that the purr is an unstable transitional state between ancestral and silent – that the data are catching snapshots of an unstable transitional state that is being replicated in several locations and populations. This should be discussed as a possibility when setting up line 562: “Our planned long-term, repeated monitoring of variation in signals and preferences across replicate Hawaiian populations ...”.

We are not exactly sure what part of Fig 1 suggests that purring is a transitional state between ancestral and silent, especially since ancestral and silent morphs predated the evolution of the purring morph (discussed in Tinghitella et al. 2018). Perhaps that the wing morphology appears somewhat intermediate? We do definitely agree, though, that we may have captured a point in time at which the frequencies of morphs within and among populations is currently unstable (an unstable transition state) and that purring shares some characteristics with each of the morphs that predate it. One of the goals of this work was to capture initial receiver responses in a rapidly evolving system. While the ratio of morphs in different populations is likely in flux, based on the data presented in this paper, we might expect purring to increase in frequency over time (unless there are other fitness costs we are not yet aware of). We now discuss the important possibility that we may have captured an unstable state in the places you suggested. The discussion now includes the following: “If we’ve captured an unstable transition state, and purring persists, as selection acts on signalers and receivers, we may see a reduction in inter-individual variation and selection favoring intended and unintended receivers that detect and locate purring crickets” (pg 20, lines 483-486) and points out the importance of long-term monitoring.

Line 148 says “If crickets respond positively to purring songs, but flies do not, this would suggest that purring is a private mode of communication amongst crickets.” Yes, and it would suggest an ESS, given current genetic variation. So, this is a crucial question that needs additional consideration in the presentation. Please address it.

Thank you for this suggestion. As we stated above, we do not feel comfortable yet claiming that purring is an ESS. However, we did add a sentence stating that purring may be an evolutionarily stable strategy (pg 16, lines 394-396).

Line 541 says “We capitalized on a unique evolutionary event ...”. Aren’t the different populations meant to be replicating independent events? Reconcile this seeming inconsistency between “unique” and replication.

We clarified our language here as suggested, referring to the evolutionary event as rare (the origin of a new signal), rather than “unique”. You are absolutely right that we see our different populations as evolving largely independently. We don’t yet know whether a single mutation to purring occurred, followed by spread to multiple locations, or if independent mutations to purring occurred in some or all locations. This is the subject of ongoing work (pg 19, line 474).

All of my concerns are conceptual and can easily be addressed in the presentation

We thank you again for your positive and constructive feedback, and we hope we have addressed your concerns in our presentation. You certainly encouraged us to think deeply about some important ideas.

Reviewer #3 (Remarks to the Author):

This is a fascinating study system and I congratulate the authors (whoever they are) on their hard work. I am well aware that conducting so many phototaxis trials per individual is a lot of effort. And the amount of work is further compounded by conducting the same phonotaxis experiments at multiple (4-6) sites. Unfortunately, at least in its present form, I do not think the study is appropriate for Nature Communications. This is not to say that it would not be well suited to a very good journal like Am Nat or Evolution.

In overview, my reasons for saying this are:

1. The writing needs to be tightened up, especially in the Introduction.

We edited the writing throughout the paper so that it is more concise, particularly in the introduction and the discussion. Even after adding all of the content suggested by reviewers, this new revised version is still 603 words shorter than the original submission. In the introduction specifically, we were able to streamline our first paragraph, remove some unnecessary details about the study system, combine and shorten sentences that covered the same content in different paragraphs, and eliminate some methodological details that are clear elsewhere in the manuscript. Edits to the discussion are outlined below.

2. I am not yet convinced that the variation in 'preferences' among individuals is due to differences in choice as opposed to activity levels at the time of capture. I know you have higher ICC for individuals than populations, but this could simply reflect that some individuals moved around (= distance + likelihood they contacted the speaker + likelihood they showed what appears to be phototaxis; I might be wrong about this last response, however, but the MS is not clear as to how you would distinguish 'positive phonotaxis' from simply moving in the general direction of the speaker).

This is an excellent point and one that we have thought about deeply. You are absolutely right that in a rectangular phonotaxis arena, you might expect for animals to bump into a speaker at one end if they are moving around a lot (and certainly some individuals are more hyperactive than others). Fortunately, we were able to measure phonotaxis behavior, specifically, in addition to metrics like distance traveled and whether or not the animals contacted the speaker. We feel confident that we are able to detect phonotaxis, specifically, because of the manner in which crickets often move when they hear and approach a sound. We identified females as phonotactic when they generally meandered (moved in a zig-zag pattern; Pollack et al. 1984; Thorson et al. 1982) in the direction of the audio stimulus without simply following the wall and without circling indefinitely. Additionally, in each trial, 2-3 trained observers had to agree upon the behavior of the animals as phonotactic or not, contacting the speaker or not, and distance travelled. We added text to the methods to clarify how we determined when individuals were positively phonotactic (now on pg 8 lines 203-206).

With exactly your concern in mind, in a subset of trials we also recorded whether or not focal females ventured >5cm from the walls of the phonotaxis arena. Of the trials in which we scored the female's behavior as positively phonotactic, 97% (224/231) of focal crickets moved >5cm away from the arena walls, while the focal animal strictly displayed thigmotaxis (wall hugging behavior) in 71% (786/1114) of trials in which we assigned non-phonotactic behavior. While this is not the only metric by which we identified positive phonotaxis, we mention this to show that there were quantitative differences in the behaviors that we observed.

It also seems worth reiterating here that in our analysis comparing responses to typical, purring, and white noise (Fig. 3) we do find evidence that phonotactic responses to purring are stronger than those to the white noise negative control that was broadcast at the same amplitude as the purrs, and that responses to typical were stronger than to purrs. These data come from trials conducted exactly as those in our phonotaxis experiments, bolstering our argument that our metrics capture phonotaxis and are capable of revealing preference. This model comparing responses to typical, purring, and white noise (Fig. 3) includes random effects at the individual-level. If our measure of phonotaxis were only representative of focal animals' motivation or activity levels, we would expect to see no differences in phonotactic responses to these categorically different stimuli.

While we are confident that the response measures that we use in the manuscript capture preference and not *only* activity levels and/or overall receptivity of individual females, we completely agree that some of the among-individual variation that we observed in our

experiments may be due to these factors. Especially in the experiments in which we only analyze responses to variants of purring song (Frequency manipulation experiment and Exemplar experiment), we agree that we need to be careful interpreting among-individual variation as purely representative of preference. We now address this excellent point in the discussion (pg 18 lines 446-452). Additionally, we carefully edited the language in the discussion to refer to female phontactic behavior as positive responses, rather than “preferences” where appropriate.

3. When it comes to the variation among individuals (Fig 4), it is not clear to me the extent to which this might simply reflect sampling error. I know you have positive values for ICC, but you do not report whether or not these differ significantly from zero. In Fig 4 it also appears that the number of females per site sampled must have been low if all individual preference functions are being drawn (except for Fig 4 c). If the lack of line is because most females did not shown phonotaxis or make contact (Fig4 a and b) (i.e. most the lines are at 0) then this suggests that part of the reason for the flatness of the population mean functions is simply due to most females not being very responsive. One line of argument that might help your case would be any evidence (perhaps from already published studies) that comparable individual preference functions for variation in ancestral calls are less variable (i.e. lower ICC) than those for purr call variation.

Thanks for these important considerations. We interpret your comment to mean that you were concerned about the sample sizes depicted in some panels of figure 4, particularly, and that those led to questions about the potential for sampling error. We did not do a good enough job explaining the makeup of Figure 4 in our first submission. The number of female crickets sampled was 114 for A - C and 37 female flies were sampled in D-E, but because there were only five purring song variants played to females, the number of possible alternative preference functions is low for binary response variables such as those presented in A, B, D, and E. Thus the lack of a line for each individual is because, where multiple females showed the same pattern of response, their preference functions overlap one another and are therefore indicated by only a single function. We have revised the figure legend to make this clear.

Nevertheless, your point about the potential sampling error is still a good one, and we've attempted to address that concern by estimating p-values for the random effects included in our mixed models (description of methods: pg 11 line 276; results: supplemental tables 1, 2, 3). The p-values mirror our original interpretation that there is significant variation in responses among individuals (or put another way, low within-individual variance, as also reflected by the ICCs), but not among populations. The only significant population random effect is for distance traveled in the exemplar experiment, but even this effect is smaller than that for individual differences (supplemental table 3).

Finally, while we really like your idea to use data from already published studies to compare variation in individual preference functions for ancestral calls to the variation in our purring call preference functions, we are unaware of any past studies with *T. oceanicus* (published or not)

that approach preference functions in a similar manner to this study. We hope that our above points, however, have allayed this fear a bit.

Specific comments

Line 53. Might there be studies of flowers that provide examples?

This is a great idea! We did search for the possibility of the evolution of a novel signal in plant communication contexts, but were unable to find any such examples. If the reviewer has a paper in mind we would love to know about it! This sentence was removed when we edited and streamlined the introduction.

Line 101. Please remind the reader when the flies reached Hawaiian islands.

We now discuss this on pg 4 line 91. The exact date of arrival for the flies is unknown, but researchers have suggested that the fly must have arrived sometime after the cricket (which was 1500 years ago or more recently; Tinghitella et al. 2011), given that there are no known alternative hosts in Hawaii. The first record of the fly in Hawaii is from 1989 (Evenhuis 2003).

Line 111. Perhaps the data is in Ref 59, but it would be good to explicitly state whether there is evidence that purring is a heritable trait and, like silent, whether it can be attributed to a single mutation.

We now explain that purring does appear to be heritable, as it has persisted in common garden for at least 12 generations (lines 100-101). Work is underway to determine whether the difference between purring and typical males can be attributed to a single mutation, like the silencing flatwing mutation, but at this point that is unknown.

Figure 1 - I would provide information for each site on:

- (a) the % of each morph present, and info as to whether this is repeatable across years
- (b) The density/abundance of the flies

Excitingly, the proportion of each morph present in each population does appear to be changing slightly with each visit to Hawaii (every 6 months). We are documenting these changes carefully and will include this information in future publications. Additionally, the distribution of morphs in our collections is not always reflective of the distribution of morphs present in each population; some morphs are difficult to capture in some locations because of burrowing behavior and other subtleties. We are addressing this challenge in current research. Thus, we hesitate to include morph proportions from just one time point in this paper. In a separate project, we (and others) have been documenting the morphological and acoustic morphs present across all populations, including new morphs that are not reported on here. As for fly abundance, because we only caught flies to the ancestral speaker (except for 1), figure 3B is an accurate representation of fly abundance across locations.

Line 140 - add a brief explanation as to why you know the sites are evolving independently.

We modified the paragraph surrounding what was line 140 during our streamlining of the introduction. We do now cite a paper that illustrates that the populations in our study are isolated with some gene flow, and together with the evidence that the presence/absence of morphs and morph frequencies differ across populations, this suggests that they are evolving independently (pg 5 lines 115). There is an in review paper using whole genome data that confirms these ideas, but unfortunately we cannot cite that until it is accepted for publication (per journal requirements).

Line 164. I know it is possible, but it does seem implausible a priori that there is a preference for novel signals. Has this ever been shown in an insect? Even the vertebrate examples in which novelty has been shown are not especially well supported cases.

It is our understanding that there are examples of preference for novel or unfamiliar signals in at least birds (e.g., Sockman et al. 2002 and 2004), fish (e.g., Farr 1977, Hughes et al. 2012; Daniel et al. 2020), and frogs (e.g., Ryan and Rand 1990) (reviewed by Rosenthal et al. 2017). We are unaware of examples in insects; nevertheless, because it is a theoretical possibility, we chose to include this possibility as one of our alternative hypotheses.

Line 202. Given you had replicates at each site, Fig 3b should include a SD or SE around the mean.

This is an excellent point, and we have added SE in Figure 3B. These error bars appear large in part due to the zero-inflated nature of our data (lots of traps capturing no flies, even at sites where flies are present).

Line 227. Personally I think it is OK, but it is worth noting that by collecting females attracted to ancestral calls, you reduce the likelihood that you then tested females that 'dislike' the ancestral call or prefer purrs. i.e. there is a selection bias.

This is an interesting point. Importantly, though, we only ever caught a single fly at a purring speaker, suggesting that flies with preferences for purring are nonexistent or at least very rare. Perhaps an ideal situation would be to capture flies using alternative methods, but we are unaware of trapping methods that do not use acoustic attraction and very little is known about the natural history of this fly, making it difficult to design something new. We now include this possibility in the discussion (pg 18 lines 455-457).

Line 308 Was it Type II or Type III ? Please confirm.

Thank you for pointing out this detail. We did in fact use Wald Type II tests to test significance of our fixed effects for the frequency manipulation experiment due to the fact that our design in this experiment did not include interaction terms. Because there are no interactions, Type II and Type III tests yield identical results. However, we did have interaction terms in the models in our

exemplar experiment and therefore used Type III tests. We had not previously made this distinction, but now make this clear on pg 12 lines 290-291.

Line 361. What was the effect size? Is this "SD" in the Supp tables? Please clarify.

This is a great question, and you are correct that we are using the magnitude of the SD of random intercepts to interpret the relative amount of variation seen within the grouping structures (individuals and populations) in our data. We have changed 'effect sizes' to 'standard deviations' on pg 13 line 331. In response to your comment above, we have also estimated p-values for our random effects which we now include in all Supplemental tables. We hope this will facilitate interpretation of our random effects.

Fig 4 Please state the sample size (number of individuals tested) for the five figures. Giving the value for each site in each figure.

We added sample sizes to figures or figure legends whenever possible (Fig 2-5).

****NB**** Line 379. Table 2 reports a pop difference in responsiveness to purrs, but Fig 3 does not appear to show this. Can you please expand on the apparent difference?

This a great observation. The apparent difference here is because Table 2 reports on data from the frequency manipulation experiment alone, while Figure 3 includes all females tested from both the frequency manipulation and exemplar experiments. We now clarify which data are reported in each Table and Figure in the table and figure legends.

Figure 5. Are the surfaces shown for multiple sites? If so, this is not apparent from the figure.

The surfaces in Figure 5 are fit to data aggregated across sites with population as a random effect within these statistical models. We have added this detail to the figure caption to improve clarity.

Discussion. I could be wrong, but I think that you need to tackle head on the concern that the extreme variation among individuals is due to sampling error. I guess the main argument against this would be to test that the ICC is significantly greater than zero (I might have missed it, but this is not reported). However, it is still not clear to me the extent to which the behavioural measures you used reflect mating preferences (as this is a one choice trial set up) rather than variation in a female's general propensity to move, which might vary for reasons unrelated to mating.

We addressed many of these concerns in our response above to your major point #2. Additionally, we added new text to the discussion where we consider the idea that broad sensory tuning, high responsiveness, and propensity to move may be important traits for the evolution of novel sexual signals. If there is high variability in the tendency to move around a lot

that may allow females to encounter and attend more closely to quiet cues (like purring) that indicate male location (and thus increase fitness). See pg 18 lines 446 - 452.

p26 Typo - no number for the Darwin reference.

Corrected. Thank you for noticing that!

Discussion: It is too long and can be shortened with no loss of value. Parts of it are too speculative. I take your point about non-significant effects being of potential biological significance. but one can equally (and more appropriately) agree that NS means either there is nothing happening or that you really need to boost your sample sizes to have enough power to detect smaller effects of biological interest.

We have dramatically streamlined the discussion, reducing its length by 515 words, and in that process eliminated discussion of nonsignificant results and much of the discussion of population level differences.

Other: Please report the bivariate correlations between distance, contact and phonotaxis. To what extent are these independent measures of preference?

We calculated these correlations separately for our Frequency manipulation experiment and Exemplar experiment and we added these values to the text (pg 8 lines 206-209 and pg 10 lines 246-248 respectively). For Frequency, the correlations of response variables are: (Phonotaxis x Contact = 0.75, Distance traveled X Phonotaxis = 0.54, Contact X Distance traveled = 0.49). For Exemplars, the correlations of response variables are: Phonotaxis x Contact = 0.7, Distance traveled X Phonotaxis = 0.51, Contact X Distance traveled = 0.44). We chose to include all three measures because we believe that each adds additional information to our understanding of receiver behavior in this case, reflecting different aspects of phonotactic behavior. For example, not all females contacted the speaker in the allotted time, yet they did exhibit phonotactic behavior. Conversely, a few very active females (as you mentioned above) managed to randomly encounter a speaker without exhibiting any behaviors suggestive of phonotactic behavior. Thus, we believe that these complementary measures provide an integrated view of female responses to song variation.

In sum, despite my reservations and calls for clarity, I do think this paper reports on a fascinating phenomenon. The core result that females respond to purrs and flies do not (yet!) does make for a clear prediction that the proportion of males purring will increase significantly in the next few years. If the authors do indeed find this is the case, then they will definitely have a great paper documenting evolution in action.

I hope my comments are useful and constructive. (And apologies for my typos - it is after midnight).

Signed: Michael Jennions

Thank you so much for your thorough feedback. It is clear that you thought deeply about our manuscript (despite the late hour :-). Your suggestions definitely improved the manuscript.

Reviewers' Comments:

Reviewer #1:

Remarks to the Author:

I read your revised version and feel like you appropriately addressed my concerns. I look forward to seeing this manuscript in print.

Susan M Bertram

Reviewer #2:

Remarks to the Author:

As before, "This is a very interesting study of an important evolutionary process in a very interesting study system. It should be of interest to a broad audience. The questions are clearly posed, the methods are sound, and the conclusions are very sensible." However, I still have some concerns about the conceptual framing.

The Introduction emphasizes that (1) this is a study system in which signals are undergoing dynamic change, (2) we should consider the manner in which the initial variation in those traits is shaped, and (3) purring might be a stable (one-dimensional) fitness peak between singing and silence.

The last sentence of the Abstract says "Our study offers a rare empirical test of the roles of natural and sexual selection in the earliest stages of signal evolution." The opening sentence of the Discussion says "We capitalized on the recent evolution of a novel song in Hawaiian populations of the field cricket *T. oceanicus* to characterize how sexual and natural selection act on new signals at their origin." I am still struggling to understand the "origin" and "the earliest stages" of purring, so that I can better understand the hypothetical evolutionary dynamics and potential stability of the trait.

The purring populations are newly discovered, but it is still unclear to me from the MS (1) how long they have existed before being discovered, how dynamically the signal morph frequency is evolving over time, and in what direction – is this dynamically changing system going from singing to purring to silent or from singing to silent to purring, or maybe both? (2) the "initial" pattern of variation is unclear – what defines "initial" here – when you discovered the purrers? The six independently evolving Hawaiian populations currently contain different ratios and combinations of the singing, purring, and silent morphs. How dynamic are these ratios? Which morph is actually the most recent ancestral morph in the six purring lineages, representing the pattern of "initial" variation in this dynamic study system?

On Kaua'i, >95% of the males are silent. Are those the ancestors to the purrers on Kaua'i, or are the ancestors from the <5% singers? If singing is ancestral, how did 95% of the population move around the unidimensional fitness peak at purring to be silent? If purring arose as a mutation after silent came to dominate, is that peak being populated by descendants of the >95% silent or descendants of the <5% singers? What about the timing of the purr mutation in the six independently evolving populations? Was the purr mutation after silent came to dominate (maybe Wailua)? Before (maybe Kapa'a)? Sometimes "just right" in the middle of the transition (La'ie – which direction)? Sometimes not yet (Hilo)?

The author's response "We are not exactly sure what part of Fig 1 suggests that purring is a transitional state between ancestral and silent, ..." still leaves my question hanging. My question is What part of Figure 1 suggests that purring is NOT a transitional state, captured in some contemporary populations that will be silent in a few more generations? In Figure 1, only Hilo seems to show a transition from singing to silent without purring possibly serving as a transitional state, but the "the absence of evidence is not evidence of absence." The experimental data on phonotaxis by female crickets and flies in Figure 3 is suggestive of a possible intermediate optimum, but is there any

data showing that individuals that purr have higher fitness than individuals that are silent? I searched the MS for "fitness" and couldn't find any evidence or claims, only cautious and apparently appropriate conditional "if"s. My concerns are also cautious and I think appropriate "What if"s.

If (1) I had a clear understanding of what evidence indicates that purring is NOT a transitional state to silence (I have no reason to believe that it is, but I'd like to see some reasons that it isn't, to eliminate the possibility, or at least to see the possibility addressed thoughtfully) and (2) I could see a clearer picture of the evolutionary dynamic of trait distributions shifting across independently evolving fitness landscapes over time, then I would be much more comfortable with the conceptual framing, with its emphasis on dynamics: initial states, early stages, dynamic change and a stable intermediate optimum. Again, all of my concerns are about the conceptual framing and can easily be addressed in the presentation.

Reviewer #3:

Remarks to the Author:

I have carefully read your responses to both my own comments and those of the other reviewers. I think you have done a very good job of revising their MS in the light of these comments. As the initially most skeptical of the reviewers, I am now pleased to say that I have revised my original conclusion that the paper was not suitable for Nat Comm. The most important change for me has been the more extensive discussion/explanation of the the extent of repeatability of the responses of individuals which reduces my concerns that high sampling error is responsible for flat preference functions.

One very minor point - I think the use of ESS on line 396 seems out of place. I know Ref 2 used this term, which is why you added it but - for what it worth - I would not given the high likelihood that you are dealing with a system which is not stable and is currently undergoing directional selection on calling.

In sum, I am now happy to support publication of the MS.

Signed: Michael Jennions

RESPONSE TO REVIEWERS

Reviewer #1 (Remarks to the Author):

I read your revised version and feel like you appropriately addressed my concerns. I look forward to seeing this manuscript in print.

Susan M Bertram

Excellent! Thank you so much!

Reviewer #2 (Remarks to the Author):

As before, "This is a very interesting study of an important evolutionary process in a very interesting study system. It should be of interest to a broad audience. The questions are clearly posed, the methods are sound, and the conclusions are very sensible." However, I still have some concerns about the conceptual framing.

The Introduction emphasizes that (1) this is a study system in which signals are undergoing dynamic change, (2) we should consider the manner in which the initial variation in those traits is shaped, and (3) purring might be a stable (one-dimensional) fitness peak between singing and silence.

Thank you for taking the time to provide further comments on our manuscript. We agree with your summary, although we do not intend to convey that purring is necessarily a stable strategy in this system (only that it may be an alternative that balances fitness from reproduction and survival in a different way from typical and silent morphs).

The last sentence of the Abstract says "Our study offers a rare empirical test of the roles of natural and sexual selection in the earliest stages of signal evolution." The opening sentence of the Discussion says "We capitalized on the recent evolution of a novel song in Hawaiian populations of the field cricket *T. oceanicus* to characterize how sexual and natural selection act on new signals at their origin." I am still struggling to understand the "origin" and "the earliest stages" of purring, so that I can better understand the hypothetical evolutionary dynamics and potential stability of the trait.

The purring populations are newly discovered, but it is still unclear to me from the MS (1) how long they have existed before being discovered, how dynamically the signal morph frequency is evolving over time, and in what direction – is this dynamically changing system going from singing to purring to silent or from singing to silent to purring, or maybe both? (2) the "initial" pattern of variation is unclear – what defines "initial" here – when you discovered the purrers? The six independently evolving Hawaiian populations currently contain different ratios and combinations of the singing, purring, and silent morphs. How dynamic are these ratios? Which morph is actually the most recent ancestral morph in the six purring lineages, representing the pattern of "initial" variation in this dynamic study system?

These are excellent points. Several of your questions are the focus of ongoing work that is beyond the scope of this manuscript (e.g. identifying the genetic background of purring individuals to help decipher whether purring arose from silent or typical morphs). However, we have clarified and added text in the Introduction (Lines 105-117) to better highlight the history, timeline, and dynamics of the morphs in the long-studied populations. Your question about what we define as "initial" is a good one. We now make clear that while it is impossible for us to know how long purring has existed in Kalaupapa (the population was composed of purring crickets when first discovered, but had not previously been studied), purring individuals were not present in other locations prior to 2017. Thus, these populations have only contained measurable numbers of purring males for less than 3 years. We describe this as "time zero", as close to the initial evolution of the trait (in these populations) as possible. We also now acknowledge that the purring morph could have existed at extremely low frequencies that were not detected (Lines 109-111).

On Kaua'i, >95% of the males are silent. Are those the ancestors to the purrers on Kaua'i, or are the ancestors from the <5% singers? If singing is ancestral, how did 95% of the population move around the unidimensional fitness

peak at purring to be silent? If purring arose as a mutation after silent came to dominate, is that peak being populated by descendants of the >95% silent or descendants of the <5% singers? What about the timing of the purr mutation in the six independently evolving populations? Was the purr mutation after silent came to dominate (maybe Wailua)? Before (maybe Kapa'a)? Sometimes "just right" in the middle of the transition (La'ie – which direction)? Sometimes not yet (Hilo)?

These are excellent questions. As we state above, future genetic studies will help to definitively show how purring morphs relate to typical and silent crickets. It is also important to note that the 95% silent and 5% typical statistics cited for Wailua were true in the early 2000s, but recent work has shown additional changes. We now refer to additional recent papers to make the dynamic situation more clear. Both Nathan Bailey's lab and our own have shown that typical males have been absent in Wailua for at least 5 years (Tinghitella et al. 2018, Pascoal et al. 2020). While far from conclusive, the appearance of purring phenotypes in Wailua is suggestive of the fact that purring originates from some modification of 'flatwing' crickets (at least within the Wailua population) allowing them to produce the novel signal. We agree with you that the origins of this novel signal (and whether these origins are consistent/shared across populations where the phenotype appears) are fascinating, but definitely answering these questions is outside of the scope of this paper.

The author's response "We are not exactly sure what part of Fig 1 suggests that purring is a transitional state between ancestral and silent, ..." still leaves my question hanging. My question is What part of Figure 1 suggests that purring is NOT a transitional state, captured in some contemporary populations that will be silent in a few more generations? In Figure 1, only Hilo seems to show a transition from signing to silent without purring possibly serving as a transitional state, but the "the absence of evidence is not evidence of absence." The experimental data on phonotaxis by female crickets and flies in Figure 3 is suggestive of a possible intermediate optimum, but is there any data showing that individuals that purr have higher fitness than individuals that are silent? I searched the MS for "fitness" and couldn't find any evidence or claims, only cautious and apparently appropriate conditional "if's". My concerns are also cautious and I think appropriate "What if's".

If (1) I had a clear understanding of what evidence indicates that purring is NOT a transitional state to silence (I have no reason to believe that it is, but I'd like to see some reasons that it isn't, to eliminate the possibility, or at least to see the possibility addressed thoughtfully) and (2) I could see a clearer picture of the evolutionary dynamic of trait distributions shifting across independently evolving fitness landscapes over time, then I would be much more comfortable with the conceptual framing, with its emphasis on dynamics: initial states, early stages, dynamic change and a stable intermediate optimum. Again, all of my concerns are about the conceptual framing and can easily be addressed in the presentation.

We appreciate your comment that "the absence of evidence is not evidence of absence", and we do agree. However, we hope we have convinced you that purring is not likely to be a transitional state. Temporally, the silent morph existed in some locations for at least 15 years before purring appeared (either by mutation or migration). This shows that silent males evolved from the ancestral morph without passing through a "purring" state. Secondly, several populations contain combinations of morphs that do not support this idea. Hilo contains only ancestral and silent (no purring). Wailua contains only purring and silent, and purring was not found there before 2018.

Reviewer #3 (Remarks to the Author):

I have carefully read your responses to both my own comments and those of the other reviewers. I think you have done a very good job of revising their MS in the light of these comments. As the initially most skeptical of the reviewers, I am now pleased to say that I have revised my original conclusion that the paper was not suitable for Nat Comm. The most important change for me has been the more extensive discussion/explanation of the the extent of repeatability of the responses of individuals which reduces my concerns that high sampling error is responsible for flat preference functions.

Thank you for the positive feedback. We worked very hard to address your thoughtful comments and believe that this version is much improved. Thank you again for the time you took to review our paper so thoroughly.

One very minor point - I think the use of ESS on line 396 seems out of place. I know Ref 2 used this term, which is why you added it but - for what it worth - I would not given the high likelihood that you are dealing with a system which is not stable and is currently undergoing directional selection on calling.

We agree and we removed the mention of ESS from the paper.

In sum, I am now happy to support publication of the MS.

Thank you again for your helpful review.

Signed: Michael Jennions